# Brief Communication: Inclusiveness in designing an early warning system for flood resilience

**Tahmina Yasmin[1], Kieran Khamis[1], Anthony Ross[2], Subir Sen[3], Anita Sharma[4], Debashish Sen[4], Sumit Sen[3], Wouter Buytaert[2], David M. Hannah[1]**

[1]School of Geography, Earth & Environmental Sciences, University of Birmingham, Birmingham, UK.
[2]Department of Civil and Environmental Engineering, Imperial College London, London, UK
[3]Centre of Excellence in Disaster Mitigation and Management, Indian Institute of Technology Roorkee, India.
[4]People's Science Institute, Dehradun, India.

**Correspondence:** Tahmina Yasmin (t.yasmin@bham.ac.uk) and David M. Hannah (d.m.hannah@bham.ac.uk)

## Abstract

Floods remain a wicked-problem and are becoming more destructive with widespread ecological-social-and-economic impacts. The problem is acute in mountainous-river-catchments where plausible-assumptions of risk-behaviour to flood-exposure-and-vulnerability are crucial. Inclusive approaches are required to design suitable flood early-warning-systems (EWS) with a focus on local social-and-governance context rather technology, as is the case with existing practice. We assess potential approaches for facilitating inclusiveness in designing EWS by integrating diverse-contexts and identifying preconditions and missing-links. We advocate the use of a SMART-approach as a checklist for good-practice to facilitate bottom-up-initiatives that benefit the community-at-risk by engaging them in every stage of the decision-making process.

## 1   Introduction

The theme for World Meteorological Day 2022 (March 23) was 'Early Warning and Early Action – Hydrometeorological and Climate Information for Disaster Risk Reduction' which emphasises the vital importance of information generation and sharing to minimize the risks from hydrometeorological extremes. Further, the United Nations secretary-general announced a major initiative, to be delivered via COP 27 (UN Climate Conference), for 'everyone on Earth should be protected by early warning systems against extreme weather and climate change within the next five years.' These policy initiatives indicate the growing need for new information and knowledge relating to risks arising directly from hazard but also from the complex interactions with exposure and vulnerability (IPCC defined risk=hazard $\times$ exposure $\times$ vulnerability $\div$ capacity to cope, see details in Cardona et al., 2012). Although our understanding of hydrological extremes, such as floods, has evolved in recent decades as we view them through the lens of hydro-complexity (Kosow et al., 2022). However, floods remain a "wicked" problem and are becoming more destructive with ecological, social and economic impacts (i.e., source of water pollution, damages to wastewater and irrigation system, excessive erosion damaging riverbank settlements, see details in Kosow et al., 2022; Hannah et al., 2020). In mountainous regions floods are becoming more unpredictable and destructive in response to increasing climatic extremes. This is exacerbated by anthropogenic pressures which have severely modified formerly pristine, high altitude river catchments. Furthermore, increased encroachment of riverbanks, dumping of solid and sewer waste and rapid urbanisation has increased the proportion of low-income communities living in flood-prone areas (Mao et al., 2018; Paul et., al., 2018). The lack of adequate hydrometeorological monitoring networks or early warning system in these regions causes undue damage to lives and property (Mountain-EVO, 2017; Pandeya et al., 2021). Yet prediction of risks associated with floods is difficult to achieve in such data-scarce mountainous regions.

Indeed, the most recent report of the Intergovernmental Panel on Climate Change (IPCC, 2022) highlighted the urgent need for investment in adaptation and resilience, particularly in developing regions which have been historically underfunded but are already impacted by extreme weather events. A key requirement is to improve early warning alerts of anticipated storms, heatwaves, floods and droughts. To generate such warning information for floods, systematic development of monitoring networks that utilise appropriate technologies are required. These systems should also consider social, cultural and political dimensions to identify context-specific understanding on inequality and its impact on assessing vulnerabilities and exposure, so that the warning system can ensure inclusiveness in responses following appropriate decision-making chains (Mao et al., 2018; Acosta-Coll et al., 2018). Such an integrated and interconnected monitoring system requires science, policy and local community-led approaches that can bring diverse stakeholders (i.e., gender, sex, age, socio-economic status and physical abilities) together and generate knowledge to guide their decision to propose solutions that fit the local context (Buytaert et al., 2018; Kosow et al., 2022; Roque et al., 2021; Zulkafli et al., 2017). Despite this call for an inclusive approach for generating an early warning alert system, the existing flood monitoring practices and designs are strongly technology-driven (i.e., information and communications technology [ICT]) and focus less on converging with the local socio-cultural and governance context (Mao et al., 2018; Westerhoff et al., 2021). There are still questions on how, where and at what level science, policy and society may converge and facilitate bottom-up initiatives for decision-making and develop innovative solutions to address challenges posed by floods.

In this commentary, we assess potential approaches for facilitating inclusiveness in the design of a flood early warning system by integrating social, cultural and political aspects, and identify preconditions and missing links.

## 2   Current approaches embedding inclusiveness in water and disaster research

In water and disaster research several approaches are emerging to provide concepts, tools and framings that can be used to support inclusiveness and disciplinary convergence for actionable knowledge production.  The concept of knowledge co-production has emerged from science-society interaction under the umbrella of adaptive governance thinking where polycentric models and power relation received attention (see details in Buytaert et al., 2018; Paul et al., 2018 and Zulkafli et al., 2017). Scholarly research has identified several potential approaches to achieve knowledge co-production under the broader umbrella of the participatory action research (PAR) including participatory modelling (Sterling et al., 2019), community-based participatory

approaches (Wallerstein et al., 2017), participatory scenario analysis (Birthisel et al., 2020; Lakhina et al., 2021; Westerhoff et al., 2021), among others. More recently, citizen science has emerged with an emphasis on "knowledge cocreation and co-generation" (i.e. the interactive processes across science, policy and implementation to collaborate and to generate knowledge for supporting environmental decision-making see further details in Buytaert et al., 2018) and new technologies, especially ICT, but limited focus on action and development. In addition, citizen science focuses more on participation by volunteers, developing trust and nurturing existing working relationships among involved actors towards knowledge co-production (Buytaert et al., 2018; Zulkafli et al., 2017).

In the contemporary disaster research literature, knowledge co-production is advocated along with participatory actions and transdisciplinary research, which laid the foundation for the participatory convergence concept to translate research into practice (Lakhina et al., 2021; Peek et al., 2020; Roque et al., 2021). Peek et al. (2020) define the participatory convergence research as 'an approach to knowledge production and action that involves diverse teams working together in novel ways—transcending disciplinary and organizational boundaries—to address vexing social, economic, environmental, and technical challenges in an effort to reduce disaster losses and promote collective well-being' (pp. 2). While this research approach has been identified as one of the best ten big ideas in funding allocation and research direction by the National Science Foundation of USA (2016), there has been little exploration on the framing (i.e., methods and ethics) to apply this in practice (Westerhoff et al., 2021). Indeed, scholars are focusing on more empirical exploration of convergence research to generate ethics and methods that may deliver successful outcomes. For example, research attempting to address coping with water extremes such as floods and droughts (Lakhina et al., 2021, Roque et al., 2021; Westerhoff et al., 2021). Recently scholars have proposed ethics that have proven useful. For example, Lakhina et al., (2021) proposed 'convergence with CARE: collaboration, accountability, responsiveness and empowerment' which require community engagement and further highlight their perspective, questions and experiences while disregarding traditional hierarchical approaches. However, much of hydrological research is focused on improving scientific measurements and developing technological solutions. For example, improving model uncertainty or the instruments and networks used to measure different facets of the hydrosphere (Beven et al., 2020) while being useful for advancing the discipline result in solutions that are often difficult to disseminate to local communities (Birthisel et al., 2020; Roque et al., 2021; Westerhoff et al., 2021). Earlier reviews indicate many empirical investigations on how social context, such as culture, politics and economics have shaped water knowledge and how and what interventions influence or shape communities' respond differently (Roque et al., 2021). This emphasises a need for future research to understand the underlying principles and ethics that would facilitate bottom-up driven activities or active participation of engaged stakeholders for knowledge co-production to responds and reshape convergence research methods.

## 3 Processes and preconditions in early warning system development

A synthesis of the literature on flood early warning systems was reviewed to develop a schematic representation of an idealised framework for developing an inclusive early warning system (Figure 1) (for more details see Acosta-Coll et al., 2018; Buytaert et al., 2018; Mashi et. al., 2020; Paul et al., 2018; Zulkafli et al., 2017). The foundation of this schematic representation (Figure 1) is adapted from the concept of knowledge co-generation processes (Buytaert et al., 2018) and co-design framing for environmental decision-making processes in a polycentric system (Zulkafli et al., 2017) and then applied with the key elements (i.e., risk knowledge; technical monitoring and warning service; communication and dissemination of warnings and community response capability (ISDR, 2020) identified by the World Meteorological Organization, International Strategy for Disaster Reduction (ISDR). All these concepts, in general advocated participatory and citizen science approach to become inclusive and generate actionable knowledge (Buytaert et al., 2018; ISDR, 2020; Paul et al., 2018; WMO, 2020). The disaster risk equation provided by the IPCC ($risk = hazard \times expousre \times vulnerability \div capacity\ to\ cope$) suggest that reduction in risk is dependent not only on efficient forecasting of hazard, but also on the understanding of associated exposure, vulnerability and capacity to cope by the exposed community. Therefore, in Figure 1, we present three interdependent steps, i.e., collate data on risk generate data and models to facilitate forecasting and disseminate that is necessary to develop a system that not

only produce flood alerts, but also provide risks information through monitoring exposure, vulnerability and capacity of the community-at-risk.

### 3.1 Mapping the risks through data collection and observation

In this step, it is crucial to collect as much information possible, to generate knowledge on the locality and the community at risk to design a purposeful early warning system. The knowledge generated can also inform on exposure, vulnerability and ability to cope if a disaster strikes and enables decision-makers to adjust or adapt necessary precautionary measures to respond efficiently in a timely manner (Buytaert et al., 2018; Pandeya et al., 2020). The required knowledge includes scientific measurements of the hydrological hazard, various context of risks information (i.e., vulnerability and exposure mapping) across the socio, cultural and political domains that contribute to the risk portfolio to be more intense and having long-term consequences (Mao et al., 2018). In general, we found most studies generate information on risk through a baseline survey of exposure and vulnerability analysis vis observation, interviews, focus group discussions, stakeholders' meetings. The data focuses on a variety of aspects including historical analysis, geographical aspects, environmental, social, economic and governance structures. All these are relevant, however, what is missing here is the lens through which it is possible to explore the complexity of the risk portfolio determined through different angles of exposure and vulnerability perceived by different stakeholders. Reaction to risks in terms of exposure and vulnerability are dependent on the social, cultural and political stances of stakeholders, and thus is highly variable (Mashi et al., 2020; Hermans et al., 2022). For instance, the communities that are living in flood vulnerable areas might not have legal rights to do so therefore, they might decide to tolerate that risk due to fear of eviction. Other stakeholders may be from state organisations which are not bound to provide services to this illegal settlement and therefore, will not engage. People might not engage also as they already lost their trust on the governance system (i.e., did not receive compensation for their previous flood damage, recurring failed commitments from the political parties to reduce flood vulnerability). Previous research partly discussed these complexities (e.g., Acosta-Coll et al., 2018; Hermans et al., 2022; Mashi et al., 2020) however, solutions to these challenges are limited.

[Figure 1]

The citizen science approach, in such cases, recommend utilising social capital tools, such as building a relationship with trust across stakeholders, identifying the people with leadership qualities or local champions (i.e., community members or a social activist/government/non-government employee who have some form of knowledge of flood risks and keen to learn about the early warning system) (Acosta-Coll et al., 2018; Mashi et al., 2020). Previous research and project experiences in a similar context demonstrated conducting structured dialogue through stakeholders' meetings, focus group discussions and forming of community groups (see further details in Acosta-Coll et al., 2018; Mashi et. al., 2020). However, these interactions can lead to confusion and unrealistic expectation relating to the monitoring system. Therefore, it is crucial to make plausible assumptions of risk behaviour relevant to flood exposure and vulnerability that can feed into designing the early warning system including having more focused conversation with the community at risks, specifying the aim and expected outcome of the flood monitoring system.

### 3.2 Forecasting hazard risks and establish an alert system in real time

This step utilises information from the previous step to identify design specifications to build the early warning system. For example, suitable sensor technology, identification of relevant variables (i.e., rainfall, water level), suitable location(s) to install the components and transmit/receive data. In addition, decision-making on data collection attributes, such as data transmission frequency, among others is critical because there will always be a trade-off between lead time and the potential for an early warning to facilitate appropriate community responses to reduce the likelihood of life. Thus, an understanding of what the optimal lead time in a certain context should be is crucial. To enable any data processing activity, adequate monitoring of relevant variables must be undertaken at the relevant spatial and temporal resolution or scale. This scale will vary depending on the topographic complexity, landcover, geology and hydrodynamic properties of the catchment of interest

(Lauden and Sponseller 2018). If historical data is limited (often the case with mountainous and logistically
challenging environments) a period of baseline data collection through the previous step is required to "get to
your catchment" before establishing a monitoring network. A range of analytical tools are available, including,
statistical modelling and simulation, to provide robust thresholds to trigger alert levels based on the collected
data. This forecasting step – i.e., predicting the likelihood of flood based on antecedent conditions - is a
challenge in data-scarce regions like the Himalaya where there may be significant uncertainty associated with
any alert/alarm thresholds due to insufficient training data (Mountain-EVO, 2017; Pandeya et al., 2019).
Therefore, many risk assumptions are involved in this step such as over-promising for a sensor-based alert
system and if the forecasts are not accurate, there may be a resentment in the community regarding the project.
This raises an important question relating to understanding the local context to get a good understanding on how
risk management happens and what this means for the design? Moreover, how and when to involve the
community (non-scientists) in the development process? Also, what is the purpose of involving the community
and other organisations and how will their involvement shape the design process? All these questions are
important for the emerging disaster risk management paradigm, where leading organisations (e.g. World
Meteorological Organisation (WMO) and other humanitarian agencies (i.e. International Federation of Red
Cross and Red Crescent Societies) are suggesting moving towards impact-based forecasting and anticipatory
humanitarian actions so that context specific risks could be identified and necessary relevant action plan could
develop on time (please see further details in report link 6).
Previous research has highlighted the importance of involving relevant state organisations, such as disaster
management departments or meteorological organisations at this stage (Acosta-Coll et al., 2018; Pandeya et al.,
2019). However, this can potentially lead to a divergence in terms of priorities; scientist and engineers are
generally focused on the success of the adopted technique and necessary data generation, while the state-led
organisations might focus on bureaucracy, policy, existing government beliefs and long-term operational plans
(e.g., maintenance and legacy costs). Therefore, engaging with the state departments at this stage can become
very difficult (Mashi et al., 2020), nonetheless from a design perspective, understanding both contexts are very
crucial for building a purposeful early warning system. The previous researcher recommended utilising a
bridging or boundary organisation that can act as a mediator and bridge the gap (Acosta-Coll et al., 2018; Mashi
et. al., 2020). Few projects involved local technological start-up companies or local research and development
organisations. However, there is limited exploration on the community engagement at this stage who struggle
to visualise such technical details in real-time application.  Further, there are also missing on the crucial aspects
of what levels of technical details to share and which is the right time/phase to share with the community or the
state authority. This inadequate understanding to decide the right time or phase will risk of over-promising for
warning alert.
*3.3 Communication and dissemination*
After installation of the alert system, identification of the best possible modes of dissemination is critical to
further interact with the vulnerable communities and communicate the potential risks along with tentative
necessary actions to minimise the risks. While this has been the most critical part, it is also one of the most
interactive components in the entire scheme. New ICT technologies such as interactive dashboard visualisations,
give more flexibility in developing the visualisation to disseminate the EWS outputs in a way that can be easily
understood by the community is a major challenge (Mashi et al., 2020; Pandeya et al., 2019). Several questions
arise in this step including a strategy to ensure the alert levels reaches to all those who are at risk, the risk
information is easy to understand and there is a desired reaction to such information. Previous research
highlights different visualisation techniques to showcase alert levels such as text, colour coding, graphics, audio
mobile messages, and showcasing locational maps (Acosta-Coll et al., 2018; Pandeya et al., 2019). What may
be missing in this step is what would be the best possible methods to communicate with the community at risk
and understanding how they perceived and responded to such forms of alerts or warnings? Here, communication
not only with the communities but also with the responsible state authorities and how they are supporting or
engaged in with the decision-making processes to respond in a timely manner.

## 4 A SMART way forward

We believe that through this commentary we have raised critical questions and identified missing links in the context of disaster resilience and the development of tools to improve preparedness and response. The most important include i) the absence of diverse contextual risk angle and community reactions; ii) a lack of community trust in government agencies and technology focused forecasting; iii) significant data limitations to ensure effective EWS operation and impact-based forecasting; and iv) a lack of effective communication strategies. All these points need deeper exploration to ensure inclusive EWS are developed in data-scarce mountainous regions or geographic regions similar in context. We acknowledge that many countries are currently implementing EWS focusing on active community participation (please see reports links 1-5) however, solutions to address these missing links are limited and thus ensuring inclusiveness and impact remained challenging. We have highlighted the need for multiple lenses to establish and explore the complexity of the risk portfolio and thus understand the architecture of the engaged stakeholders and their behaviour. This is essential to ensure actionable knowledge is generated and bottom-up initiatives are strengthened and the capacity to respond is improved.

Based on the above discussions of key questions, missing links and design needs, we propose the '**SMART** convergence participatory research' approach to support the EWS development phase and provide a checklist of good practices. The SMART approach highlights crucial activity layers to incorporate into EWS development which can help guide multi-disciplinary teams (e.g. disaster risk manager, hydrologist, engineer, and social scientist) (Figure 2). This will enable to incorporate diverse disciplinary lenses (i.e., social science and meteorological data) along with risks diversity identify by the community-at-risk (illegal settlement beside riverbank or slums) which mentioned earlier as missing-link. This will support to expose vulnerability and risks from different socio-cultural, institutional and scientific context. Following a **SMART** approach will ensure inclusiveness by helping to identify and connect missing components and linkages when designing an EWS.

The first step, **S,** represents '**S**hared understanding of the risks' ensuring all stakeholder engagements are diverse and representative (irrespective to their gender, sex, age, socio-economic status and physical abilities) and a wide range of data forms and collection methods are utilised, as stated in EWS step-1 (Figure 1). This knowledge generated from the community will help the expert group to better understand context specific risks with more focused exposure and vulnerability analysis. This further helps to identify common goals and anticipate damage from the natural hazards and thus ensures impact though appropriate forecasting.

Secondly, **M** representing '**M**onitoring of the risks' aligned closely with establishing alert system and forecasting hazard information as stated in step-2 (Figure 1). This includes an intersection of generated knowledge that will lead towards practicing collaborative activities, such as trust-building (which is key to inclusive and impact-based forecasting), exchanging critical risk information to enrich data sets, feedbacks, forming small groups for maintaining forecasting system.

Thirdly, **A**, building **A**wareness (i.e., training and capacity development activities to embed understanding of real time weather and alert information) is critical for this approach and is a continuous process throughout the development and utilisation of early warning system, in particular focus to EWS step 3 to support effective communication and dissemination and will further also support legacy and sustainability of the warning system into the local context.

Finally, **RT** indicating pre-planning **R**esponse actions on **T**ime (i.e., comprehensive disaster management plan, evacuation plan) based on the alert produced by the EWS and could be used to inform the effectiveness of the overall EWS to minimize risks from the anticipated hazard. This will inform further the level of knowledge produced through collaboration and how this can facilitate effective action by the community and responsible agencies.

[Figure 2]


We advocate the use of this **SMART** approach to facilitate bottom-up initiatives for developing an inclusive
and purposeful early warning system and to benefit the community-at-risk by engaging them every step of the
way along with including other stakeholders at multiple scales of operations (i.e., scientific and policy actors).
We advocate that the **SMART** convergence approach along with the dominant largely top-down initiatives will
contribute to developing capacity and redefining adaptation and resilience in the face of more extreme water
extremes (floods, droughts) and increased uncertainty under global change.

**Figures (1& 2)**

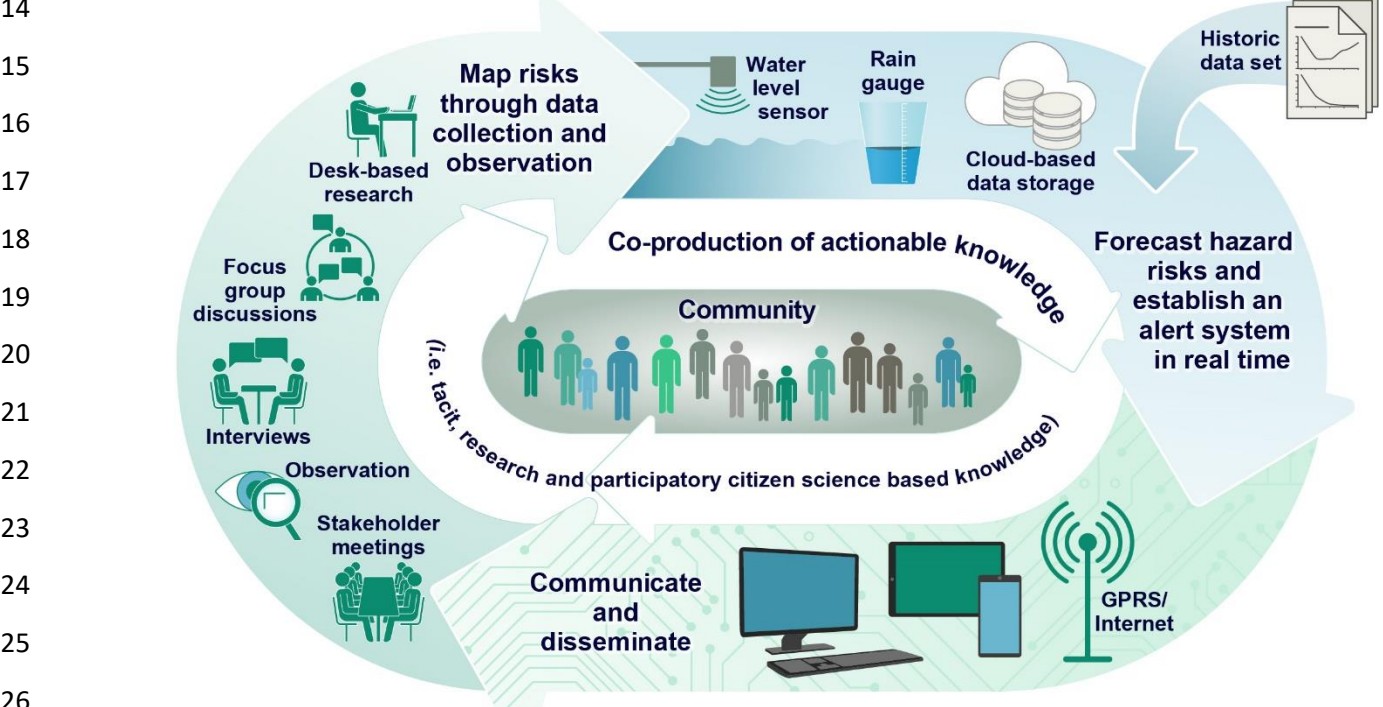

*Figure 1: An idealised scenario for developing a monitoring and alert system to provide an early
warning of potentially life/livelihood threatening natural hazards.*


Figure 2:

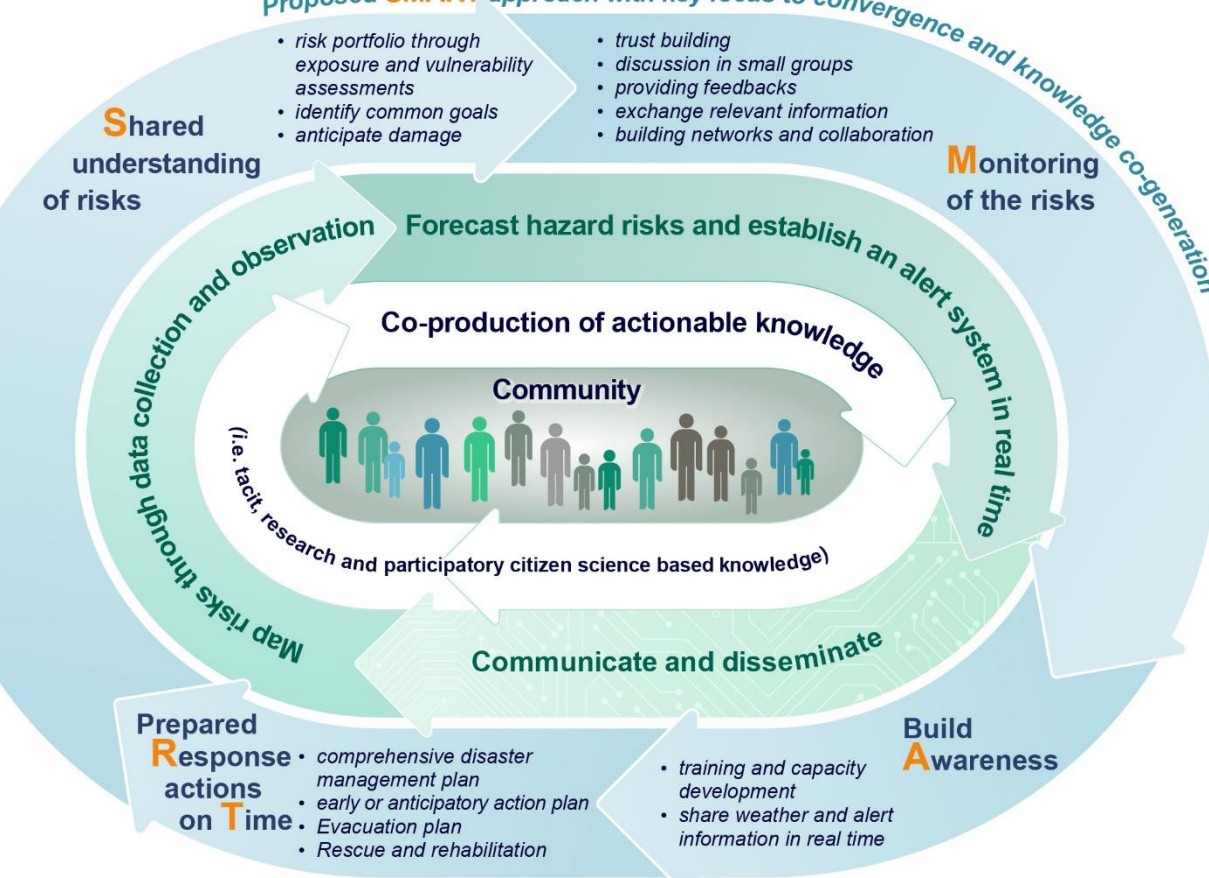


*Figure 2: A SMART convergence research approach to ensure inclusiveness in designing monitoring and alert system to provide early warning information to minimize disaster risks.*














## Authors contribution

TY and DMH prepared the manuscript with contributions from all co-authors.

## Acknowledgement

The research was funded by the Natural Environment Research Council (UKRI NERC), Research grant reference no: NERC COP26 A&R Project Scoping Call -2021COPA&R31Hannah.

## Useful links and report links

1. Mountain-EVO (2017) project paper and reports are available at: paramo.cc.ic.ac.uk/espa/
2. International Strategy for Disaster Reduction (ISDR). Emerging Challenges for Early Warning Systems in context of Climate Change and Urbanization. Available online: http://www.preventionweb.net/files/15689_ewsincontextofccandurbanization.pdf
3. Guidelines on Early Warning Systems and Application of Nowcasting and Warning Operations (2010) by World Meteorological Organization (p. 25) and can be access through https://library.wmo.int/doc_num.php?explnum_id=9456
4. Explained: Why India's Early Warning Systems For Floods And Cyclones Fall Short (indiaspend.com)
5. Community-Based Flood Early-Warning system-India: A collaborative project by the International Centre for Integrated Mountain Development (ICIMOD), Aranyak and Sustainable Eco Engineering (SEE) https://unfccc.int/climate-action/un-global-climate-action-awards/winning-projects/activity-database/community-based-flood-early-warning-system-india?gclid=Cj0KCQjw--2aBhD5ARIsALiRlwBy8J63opnqOTpqi_9ciM31ONeEat2vk2S1bNk88d-IfxpVYIpld1MaAkpeEALw_wcB
6. The Future of Forecasts: Impact-Based Forecasting for Early Action: a joint report by the Red Cross Red Crescent and the UK Met Office, https://www.anticipation-hub.org/download/file-58;
7. World Meteorological Organization (WMO) Guidelines on Multi-hazard Impact-based Forecast and Warning Services https://library.wmo.int/?lvl=notice_display&id=21994#.YvN5LnbMKUk.

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
