# Peer review of "Brief Communication: Inclusiveness in designing an early warning system for flood resilience"

_EGUsphere, 2022_

## Author Response (AR1)

Dear Editor and reviewers,

Thank you for your comments and suggestions. In response to overall comments, we have significantly improved our manuscript to clearly demonstrate the proposed SMART approach and it's added value to ensure inclusiveness in developing monitoring and alert system to provide early warning information in particular context to flood risk management. We also included our responses to the reviews' comments with indicating necessary revision and corrections in the revised manuscript. Please see the supplement documents as our responses to individual comments. We have added the lines in red and italic into the paragraph to make it clearer and specific in response to your comments.

We really appreciate your time and effort to review our paper and providing very structured comments.

Regards,

Tahmina Yasmin and David M. Hannah (On behalf of all co-authors).

**Authors responses to reviewer 1 comments**

| Reviewer 1' comments | Authors response |
|---|---|
| | We would like to thank both of our anonymous reviewers for their constructive comments and suggestions. We have provided our response and carefully addressed the issues raised by the reviewers. |
| Reviewer 1: General comments | Authors response |
| 1. This paper provides a good overview of the components of early warning systems, identifies some gaps and provides recommendations how early warning systems can become more inclusive. The paper would benefit from clearly defining how inclusion/ inclusiveness is interpreted in the context of this research. | Thank you for your comment. We altered the text and written in page 1 (line 76-88) to define more clearly what is meant by "inclusion" in this paper.

*'To generate such warning information for floods, systematic development of monitoring networks that utilise appropriate technologies are required. These systems should also consider social, cultural and political dimensions to ensure responses following appropriate decision-making chains (Mao et al., 2018; Acosta-Coll et al., 2018). Such an integrated and interconnected monitoring system requires science, policy and local community-led approaches that can bring engaged stakeholders together and generate knowledge to guide their decision to propose solutions that fit the local context (Buytaert et al., 2018; Kosow et al., 2022; Roque et al., 2021; Zulkafli et al., 2017). Despite this call for an inclusive approach for generating early warning alert system, the existing flood monitoring practices and designs are strongly technology-driven (i.e., information and communications technology [ICT]) and focus less on converging local socio-cultural and governance context (Mao et al., 2018;* |

| | |
|---|---|
| | *Westerhoff et al., 2021). There are still questions on how, where and at what level science, policy and society may converge and facilitate bottom-up initiatives for decision-making and develop innovative solutions to address challenges posed by floods.'*

We have added the lines in red into this paragraph to make it clearer and more specific:

*'To generate such warning information for floods, systematic development of monitoring networks that utilise appropriate technologies are required. These systems should also consider social, cultural and political dimensions to identify context-specific understanding on inequality and its impacts on assessing vulnerabilities and exposure, so that it can ensure inclusiveness in responses following appropriate decision-making chains (Mao et al., 2018; Acosta-Coll et al., 2018). Such an integrated and interconnected monitoring system requires science, policy and local community-led approaches that can bring  diverse stakeholders (i.e., gender, sex, age, socio-economic status and physical abilities) together and generate knowledge to guide their decision to propose solutions that fit the local context (Buytaert et al., 2018; Kosow et al., 2022; Roque et al., 2021; Zulkafli et al., 2017). Despite this call for an inclusive approach for generating early warning alert system, the existing flood monitoring practices and designs are strongly technology-driven (i.e., information and communications technology [ICT]) and focus less on converging local socio-cultural and governance context (Mao et al., 2018; Westerhoff et al., 2021). There are still questions on how, where and at what level science, policy and society may converge and facilitate bottom-up initiatives for decision-making and develop innovative solutions to address challenges posed by floods.'* |
| 2. The paper advocates community engagement along each step, but it does not disaggregate the community and discuss how the unique capacities and needs of different (marginalised) groups such as women, girls, children, persons with disabilities, elderly and illiterate, need to be considered, engaged and utilised to make it more inclusive. This aspect should be recognised throughout the paper and especially within the SMART approach. | Thank you for pointing this out – we have now closely reviewed and subsequently revised line where its relevant. Please see author response to comment 1 and 9. |
| | Specific comments |
| 3. The abstract would benefit from succinctly explaining the gap around inclusion in EWS and providing more details on how SMART fills the gap. | Please see the below revision in the Abstract.

*Floods remain a wicked-problem and are becoming more destructive with widespread ecological-social-and-economic impacts. The problem is acute in mountainous-river-catchments where plausible-assumptions of risk-behaviour to flood-exposure-and-vulnerability are crucial. Inclusive approaches are required to design suitable flood early-warning-systems (ESW) with a focus on local social-and-governance context rather technology, as is the case* |

| | |
|---|---|
| | *with existing practice. We assess potential approaches for facilitating inclusiveness in designing EWS by integrating diverse-contexts and identifying preconditions and missing-links. We advocate the use of a SMART-approach as a checklist for good-practice to facilitate bottom-up-initiatives that benefit the community-at-risk by engaging them in every stage of the decision-making process.* |
| 4. In Section 2, suggest drawing on literature/ experience around 'local knowledge' capacities of the communities and how a truly inclusive or co-produced EWS will utilise this knowledge e.g., Hermans et al. 2022 (link: https://pure.iiasa.ac.at/id/eprint/18112/1/ Hermans2022_Article_ExploringTheIntegrationOfLocal.pdf). Furthermore, the paper does not draw on Community-based EWS literature and practical experiences, for example Macherera and Chimbari, 2016 (Link: https://www.ncbi.nlm.nih.gov/pmc/articles/PMC6014131/). | Our brief communication focused on inclusiveness in designing an early warning system for flood and section 2 largely discuss approaches in water and disaster research to become inclusive in designing early warning system that build upon knowledge co-production and convergence platform where diverse and context-specific community knowledge intersects with knowledge derived from disciplinary experts. Therefore, we focus on identifying underlying principles and ethics for designing EWS. While the Macherera and Chimbari, 2016 review paper is focused on community-based early warning systems for human diseases; therefore, it is not highly relevant to this short communication. However, we will include Hermans et al., 2022 in the revised paper as reference and we will revise associated text accordingly. |
| 5. Additionally, suggest explaining 'knowledge co-generation' already in section 2 seeing as this is the foundation for the framework. | We have now added explanation on 'knowledge co-generation' in section 2, page 3 (line 103-106) – making reference to previous literature for more in-depth explanation

*'More recently, citizen science has emerged and emphasises on "knowledge cocreation and co-generation" (refers to the interactive processes across science, policy and implementation to generate knowledge for supporting environmental decision-making and is adopted from two distinct paradigms: (1) science-society interaction and (2) collaborative knowledge production, see further details in Buytaert et al., 2018) with limited focus on action and development but more on new technologies, especially ICT.'* |
| 6. In Section 3.1/2, suggest including some discussion/reference around impact-based forecasting (IBF) which focuses on generating information on what the weather will 'do' (by fusing exposure and vulnerability information with hazard forecast info) instead of what the weather will 'be' (traditional forecast). You can find guidelines from WMO (https://library.wmo.int/?lvl=notice_display&id=21994#.YvN5LnbMKUk ) and the Met Office/ Climate Centre (link: https://www.anticipation-hub.org/download/file-58 ). IBF strongly advocates for partnership and multi-stakeholder engagement. | Thank you for your suggestion. We have now added lines on page 5 line (215-221).

*'All these questions are also important for emerging disaster risk management paradigm where leading humanitarian organisations (i.e., World Meteorological Organisation (WMO), International Federation of Red Cross and Red Crescent Societies) are suggesting moving towards impact-based forecasting and anticipatory humanitarian actions so that context specific risks could be identified and necessary relevant action plan could develop on time (please see further details in https://www.anticipation-hub.org/download/file-58; https://library.wmo.int/?lvl=notice_display&id=21994#.YvN5LnbMKUk).'* |
| 7. Section 3.3 could draw on experiences of understanding communities' preferences for different warning communication technologies and | Thank you for your comment. Actually, previous literature covers this in very traditional ways, as we mentioned in, page 5, line 245-253. Therefore, through our commentary we have |

| | |
|---|---|
| designing the format of the message to ensure it is understandable and actionable by different groups in the community. See for example Cumiskey et al. 2015 (link: https://www.emerald.com/insight/content/doi/10.1108/IJDRBE-08-2014-0062/full/html) | raised that it should need more consultation and understanding of the context and need to focus on what community along with other responsible authorities prefer as for communicating alert messages.

*'Several questions arise in this step including a strategy to ensure the alert levels reaches to all those who are at risk, the risk information is easy to understand and there is a desired reaction to such information. Previous research highlights different visualisation techniques to showcase alert levels such as text, colour coding, graphics, audio mobile messages, and showcasing locational maps (Acosta-Coll et al., 2018; Pandeya et al., 2019). What may be missing in this step is what would be the best possible methods to communicate with the community at risk and understanding how they perceived and responded to such forms of alerts or warnings? Here, communication not only with the communities but also with the responsible state authorities and how they are supporting or involving with the decision-making processes to respond in a timely manner.'* |
| 8. Figure 1 does provide a good overview of the different stages in an EWS but it is missing details on how each component can be inclusive apart from broadly showing that the community should be engaged in each step. The pictures within the diagram could refer to specific inclusive approaches/ activities e.g., activities like engaging women or schools in water level data collection, involving them in the risk assessments design of early action plans, working with local leaders, disseminating warnings in multiple ways to reach different groups. Some of these approaches are mentioned in the text for the SMART approach but I'm still missing a clear overview of all the tools/ guide on how to actually realise the 'co-production of actionable knowledge'. | Figure 1 represent a schematic of an idealised EWS based on literature review - with particular attention towards flood early warning. The various participatory techniques, such as stakeholder meetings, interviews, focus group discussion includes diverse stakeholder to be inclusive. However, as the descriptions of the three steps in this schematic further focus on the gaps and questions that need answer to become inclusive. Based on these discussions later in this commentary we proposed SMART approach as a checklist for good practice and a layer to add with this EWS developmental steps so that it could become inclusive. In the revised version we added few lines in the proposed SMART approach (please see response to comment 9) to make it clearer on how this will ensure inclusiveness in designing EWS. |
| 9. The title of Figure 2 specifies inclusiveness in disaster risk management not early warning systems. Many components of the figure are not well explained in the short text. In my opinion for the purposes of this paper, it would be more useful to expand the details of the SMART approach and focus on inclusiveness for EWS rather than half of the figure being about the top-down approach with the overall goal for redefining adaptation and resilience rather than inclusive EWS. | We have now revised figure 2 and rewritten section 4 in page 6 (line 262-279).

We highlight crucial steps for multi-disciplinary team (disaster risk manager, hydrologist, engineer, and social scientist) to follow when exploring risk architectures and planning response actions (Figure 2). These include Firstly, S representing 'Shared understanding of the risks' providing a scope for including diverse stakeholder engagements (irrespective to their gender, sex, age, socio-economic status and physical abilities) in different data collection as stated in step-1 (Figure 1). This knowledge generated from the community will help the expert group to better understand context specific risks with more focused portfolio to map out risks' factors through exposure and vulnerability analysis. This further helps to identify common goals and anticipate damage from the natural hazards. Secondly, M representing 'Monitoring of the risks' aligned closely with establishing alert system and forecasting hazard information as stated in step-2 (Figure 1). This includes an intersection of generated knowledge that will lead towards practicing collaborative activities, such as through |

| | |
|---|---|
| | knowledge co-production and collaboration (i.e., trust-building, exchanging critical risk information, providing feedbacks, forming small groups for maintaining forecasting system. Thirdly, **A** as in building **A**wareness (i.e., training and capacity development activities, understanding weather and alert information in real time) is critical for this approach and is a continuous process throughout the development and utilisation of early warning system. Finally, **RT** indicating and pre-planning **R**esponse actions on Time (i.e., comprehensive disaster management plan, evacuation plan) is crucial to minimize risks from the anticipated damages from the hazard information and will inform the existing community and responsible agencies to take effective action. |
| 10. An option could be to merge the SMART component of Figure 2 into an expanded Figure 1. This way one could just focus what the SMART approach tangibly means for each component/step of the EWS to make it inclusive. Having one useful figure to explain inclusion in EWS and the SMART approach would elevate the value of the paper. | Thank you for your suggestion. We have revised Figure 2 to make it clearer and more specific to our paper objective. We have highlighted and added description in figure 2. Please see responses to comment 9 and 14 |
| 11. The title of Figure 1 using 'natural disasters' should be changed to 'natural hazards'. | Thank you. Revised accordingly. |
| 12. The SMART approach specifies 'response actions' but if these are taken ahead of the impact of the hazard then these should be 'early or anticipatory actions' as implemented by NGOs, Red Cross Red Crescent and UN agencies. Tozier de la Poterie (2021) provides more insights into anticipatory action planning which may be useful for the authors to explore (link: https://www.tandfonline.com/doi/full/10.1080/17565529.2021.1927659). There is also growing interest into making anticipatory action programmes within the humanitarian sector more inclusive. See for example FAO, 2020 (link: https://www.fao.org/3/cb1072en/cb1072en.pdf), and there is a dedicated related protection, gender and inclusion resource page on the Anticipation Hub (Link: https://www.anticipation-hub.org/learn/emerging-topics/protection-gender-and-inclusion-inanticipatory-action) which may be of interest to the authors. | Thank you for your suggestion. We have revised and added a few lines on this. Please see author response in comment 6. |
| Technical corrections | |
| 13. There are several typos and grammatical errors in the paper. I have noted some of these below, but this list is not exhaustive and suggest that the authors thoroughly check the paper for errors to improve the readability of the paper.:
 Title – use either 'an early warning system' or 'early warning systems'
 Abstract: Communities-at-risks – remove 's' after risk | Thank you for these corrections. Revised and corrected accordingly. |

| | |
|---|---|
| Introduction: Live and property (page 2 line 70) – add 's' to live
Page 2 line 75 – historically underfunding to 'underfunded'
Page 2 line - line 83 'an' early warning alert system, line 85 'with the' local…. 89/90 – add
'a' flood early warning system, line 91 title – suggestion 'current approaches facilitating'
Page 3 – line 103 add 'ships' to working relation
Page 5 – A SMART 'way forward; 253 'involving with' change to 'engaged in' | |

**Authors responses to reviewer 2 comments**

| Reviewer 2 comments | Authors response |
|---|---|
| | We would like to thank both of our anonymous reviewers for their constructive comments and suggestions. We have provided our response and carefully addressed the issues raised by the reviewers. |
| 1. The manuscript on 'Inclusiveness in designing early warning system for flood resilience' by Yasmin et al is interesting. However, still the message of the short communication is not clear to me. I would suggest major revision taking into account the following comments:
1) The authors should provide a clear description how the SMART approach resolves existing limitations of flood warning system. | We have now revised figure 2 and rewritten section 4 in page 6 (line 262-279).

*We believe that through this commentary we have raised critical questions and identified missing links in the context of disaster resilience and the development of tools to improve preparedness and response. The most important include i) the absence of diverse contextual risk angle and community reactions; ii) a lack of community trust in government agencies and technology focused forecasting; iii) significant data limitations to ensure effective EWS operation and impact-based forecasting; and iv) a lack of effective communication strategies. All these points need deeper exploration to ensure inclusive EWS are developed in data-scarce mountainous regions or geographic regions similar in context. We acknowledge that many countries are currently implementing EWS focusing on active community participation (please see reports links 1-5) however, solutions to address these missing links are limited and thus ensuring inclusiveness and impact remained challenging. We have highlighted the need for multiple lenses to establish and explore the complexity of the risk portfolio and thus understand the architecture of the engaged stakeholders and their behaviour. This is essential to ensure actionable knowledge is generated and bottom-up initiatives are strengthened and the capacity to respond is improved.* |

*Based on the above discussions of key questions, missing links and design needs, we propose the 'SMART convergence participatory research' approach to support the EWS development phase and provide a checklist of good practices. The SMART approach highlights crucial activity layers to incorporate into EWS development which can help guide multi-disciplinary teams (e.g. disaster risk manager, hydrologist, engineer, and social scientist) (Figure 2). This will enable to incorporate diverse disciplinary lenses (i.e., social science and meteorological data) along with risks diversity identify by the community-at-risk (illegal settlement beside riverbank or slums) which mentioned earlier as missing-link. This will support to expose vulnerability and risks from different socio-cultural, institutional and scientific context. Following a SMART approach will ensure inclusiveness by helping to identify and connect missing components and linkages when designing an EWS.*

*The first step, S, represents 'Shared understanding of the risks' ensuring all stakeholder engagements are diverse and representative (irrespective to their gender, sex, age, socio-economic status and physical abilities) and a wide range of data forms and collection methods are utilised, as stated in EWS step-1 (Figure 1). This knowledge generated from the community will help the expert group to better understand context specific risks with more focused exposure and vulnerability analysis. This further helps to identify common goals and anticipate damage from the natural hazards and thus ensures impact though appropriate forecasting.*

*Secondly, M representing 'Monitoring of the risks' aligned closely with establishing alert system and forecasting hazard information as stated in step-2 (Figure 1). This includes an intersection of generated knowledge that will lead towards practicing collaborative activities, such as trust-building (which is key to inclusive and impact-based forecasting), exchanging critical risk information to enrich data sets, feedbacks, forming small groups for maintaining forecasting system.*

*Thirdly, A, building Awareness (i.e., training and capacity development activities to embed understanding of real time weather and alert information) is critical for this approach and is a continuous process throughout the development and utilisation of early warning system, in particular focus to EWS step 3 to support effective communication and dissemination and will further also support legacy and sustainability of the warning system into the local context.*

*Finally, RT indicating pre-planning Response actions on Time (i.e., comprehensive disaster management plan, evacuation plan) based on the alert produced by the EWS and could be used to inform the effectiveness of the overall EWS to minimize risks from the anticipated hazard. This will inform further the level of knowledge produced through collaboration and how this can facilitate effective action by the community and responsible agencies.*

| 2) The authors should provide evidences and examples of the application of SMART approach. | In paper we propose SMART approach as a way to answer the questions that we have raised by reviewing each EWS steps that developed from reviewing flood early warning related literature.

This article seeks to outline the concept of the SMART approach as a brief communication. It is not a data paper and therefore giving examples seems beyond the scope of this article type. We make reference to the literature by way of evidencing approaches throughout the manuscript. |